# Model-Agnostic Social Network Refinement with Diffusion Models for Robust Social Recommendation

## Abstract

Social recommendations (SRs) aim to enhance preference modeling by integrating social networks. However, their effectiveness is mainly constrained by two factors: the noisy social connections that may not reflect shared interests, and the limited number of social connections for most users, which hampers the system's ability to fully leverage social influence. Therefore, it is essential to perform social network refinement by removing noisy connections and adding meaningful ones for robust SRs. Inspired by the denoising capability of generative diffusion models, we propose a Model-Agnostic Social Network Refinement framework with Diffusion Models for Robust Social Recommendation (ARD-SR). Specifically, in the forward process, we corrupt the social network by progressively adding position-specific Gaussian noise calibrated to the user preference similarity, better simulating how the social network responds to noise perturbations. The reverse process learns to denoise, guided by each user's neighborhood preferences from the SR backbone, generating a tailored social network aligned with each user's preference for establishing connections. For effective learning, we design a curriculum-based training mechanism that progressively introduces challenging samples characterized by high sparsity or high noise levels. Finally, ARD-SR and the SR backbone are alternately trained, ensuring a continuous mutual enhancement between the social network refinement and the backbone's user representation learning. To further enhance the quality of the refined social network, (1) we introduce a preference-guided flip operation during inference to improve the input quality; and (2) we modify social connections based on the exponential weighted moving average of ARD-SR's predictions across epochs to reduce fluctuations. Experiments on three datasets show that ARD-SR significantly improves SR performance across multiple SR backbones.

## 1 Introduction

Social recommendations (SRs) aim to model users' social neighbor influence to enhance preference learning, following the homophily theory [22], i.e., socially connected users tend to share similar interests. It effectively mitigates the data sparsity issue and has seen notable achievements in improving recommendation accuracy, especially with advancements in deep learning technologies such as graph neural networks [3, 39] and self-supervised learning [15, 47]. However, social networks are inherently noisy, as connections between users do not always indicate similar interests [13, 28]. The indiscriminate use of all social connections, as seen in most existing work [2, 6, 29, 46], can hinder user preference modeling. Additionally, many users have few or no social neighbors, limiting the potential of leveraging social influence to capture their interests. Thus, it is necessary to perform social network refinement by removing noisy connections and adding meaningful ones, thereby constructing a high-quality social network to facilitate robust SRs.

Social network refinement can be framed as a link prediction task, where connection likelihoods between users are estimated to guide the edge removal or addition. Existing approaches can be categorized into heuristic-based and model-based methods. The former [13, 26, 45], relies on heuristics, such as user preference similarity, to add or remove edges. However, the absence of supervision from social connection labels limits the alignment of preference similarity with actual social connections. The latter, in contrast, trains link prediction models with supervised [23] or self-supervised signals [28, 41] derived from social networks. Nonetheless, the inherent noise in the social network may compromise the models' robustness. Besides, they fail to add useful edges and predict connections based only on pairwise user features, overlooking the broader dependencies among users in the entire social network.

Recently, diffusion models (DMs) have achieved state-of-the-art performance in image generation tasks [1, 10, 27]. DMs gradually corrupt the input with small amounts of random noise during the forward process and then learn to recover it step by step in the reverse process. Compared to other generative models, such as VAEs [16] and GANs [4], DMs offer superior training stability and are not susceptible to model collapse. DMs' denoising characteristics and generative nature align well with obtaining high-quality social networks for robust SRs, providing several benefits. First, DMs' intrinsic denoising ability enables the recovery of clean social networks from noisy ones. By decomposing the denoising process into numerous incremental steps, DMs simplify the overall task and improve robustness to noisy inputs. Second, DMs offer flexible generation capabilities conditioned on specific inputs [1, 11], allowing for personalized generation even when a user has sparse or no observed social connections. Moreover, DMs can comprehensively capture the underlying data distribution, better capturing global dependencies in the whole social network.

However, unlike images, where individual pixels lack explicit semantic meaning, social networks are represented as adjacency matrices, with each entry signifying a meaningful connection between users. Furthermore, image pixels exhibit local dependencies within a grid structure, while social connections are not confined by spatial proximity, allowing nodes to link non-locally and form long-range dependencies. Additionally, image pixels are continuous values, while the binary social adjacency matrix is highly sparse, with most entries being zero. In light of this, two main challenges are confronted: (1) How can DMs be adapted to better capture

the semantics and long-range dependencies inherent in social networks? (2) How to achieve effective network refinement, given the predominance of zero values in the social adjacency matrix?

To address these challenges, we propose ARD-SR, a model-agnostic social network refinement framework based on DMs, designed to integrate seamlessly with any SR backbone for robust SRs. Technically, ARD-SR progressively corrupts the social adjacency matrix with position-specific noise in the forward process, assigning more noise to users with lower preference similarity. In the reverse process, ARD-SR learns to denoise the corrupted samples, guided by the user's neighbor preferences learned from the SR backbone, providing personalized context to steer denoising. To facilitate effective learning, we devise a curriculum-based training mechanism, initially focusing on simpler samples—those users with more neighbors and less noise—and gradually introducing more complex ones. ARD-SR is then alternately trained with the SR backbone for continuous mutual enhancement: the refined social network from ARD-SR helps calibrate user representations in the SR backbone, which in turn better informs the position-specific noise schedule and guides the reverse denoising process in ARD-SR. To further improve the quality of the refined social network, we (1) introduce a preference-guided flip operation to correct the input of the inference process, thus alleviating the input sparsity issue and introducing useful social signals for improved generation and (2) modify edges based on the exponential weighted moving average of ARD-SR's predictions over epochs to smooth out prediction fluctuations.

The main contributions of this paper are three-fold. **(1)** We are the first to propose a model-agnostic social network refinement framework, which exploits the denoising and generative capability of DMs to enhance the robustness of any existing SR backbone. **(2)** For more effective social network refinement with DMs, we introduce a position-specific noise schedule in the forward process and guide the reverse process with neighborhood preferences, enabling robust and personalized social network refinement. We further design a curriculum-based training mechanism to gradually introduce challenging samples, for more effective learning. **(3)** We integrate ARD-SR into multiple SR backbones, validating its efficacy in enhancing SR performance across three real-world datasets.

## 2 Related Work

**Social Recommendation (SR)**. SR aims to enhance preference inference by modeling social influence within the social network. Early SR methods are mainly matrix-factorization (MF) based. For example, SoReg [21] and CNSR [38] align representations of socially connected users by adding regularization terms to the MF loss; TrustMF [40] and TrustSVD [6] co-factorize user-item interactions and the social network via shared user embeddings. Recent advances have introduced graph neural networks (GNNs) and self-supervised learning (SSL) approaches, where GNNs allow SRs to capture complex interdependencies within social networks such as GraphRec [2], DiffNet++ [37], DMJP [29] and DSR [25], while SSL creates auxiliary tasks that improve user representation learning, such as MHCN [46], SEPT [44] and DSL [33].

**Social Network Refinement for Robust SR**. Social network refinement seeks to improve network quality by removing noisy edges that impede user preference modeling and adding supportive edges to benefit users with sparse connections, thereby enhancing the robustness of SR. For example, heuristic-based methods perform edge removal and addition based on preference similarity, as in ESRF [43] and SHaRe [13]. However, without supervision from social connection labels, preference similarity may not accurately align with true social connections. Model-based methods like GDMSR [23] train link prediction models to remove noisy edges; however, noise within the social network can compromise the reliability of supervision signals. Recently, self-supervised methods like SSD-ICGA [28] and GBSR [41] use contrastive learning to identify noisy social signals through dropout-based augmentation. However, they fail to insert new edges. Moreover, they primarily assess edge relevance based on pairwise user relationships, potentially overlooking broader global dependencies within the social network.

**Diffusion Models (DMs) for Recommendation**. DMs have become a powerful tool for generative tasks, first popularized by DDPM [10]. Subsequent advancements like sampling efficiency [27] and conditional diffusion [1, 11] have made DMs competitive with VAEs [16] and GANs [4], without model collapse or training instability issue. Inspired by this, some efforts have been devoted to integrating DMs into recommendation [20]. They are applied either in the graph space, to generate user-item interactions as in DiffRec [35] and CODIGEM [32], or in the latent space to generate user/item embeddings such as DiffKG [14], DreamRec [42] and DDRM [48]. Recently, RecDiff [19] uses the DM for denoising in the latent social space but noisy edges still participate in the representation learning, thus the effectiveness in mitigating noise is not assured; GDSSL [18] trains a DM to directly denoise the social network. However, the training of DM is not end-to-end with the recommendation task, and it only drops noisy edges without adding new ones. In contrast, we propose a model-agnostic social network refinement approach (removing and adding connections) based on DMs, trained end-to-end with the SR backbone, for robust SR.

## 3 Preliminaries

### 3.1 Notations and Problem Statement

**Notations.** We consider a set of users $\mathcal{U} = \{u_1, \ldots, u_m\}$ and items $\mathcal{I} = \{i_1, \ldots, i_n\}$. Users exhibit two behaviors: consuming items or connecting with other users, represented by the user-item interaction graph $\mathcal{G}_R$ and the user-user social graph $\mathcal{G}_S$. These graphs correspond to the binary adjacency matrices $\mathbf{R} = \{r_{ui}\}_{m \times n}$ and $\mathbf{S} = \{s_{uv}\}_{m \times m}$, where $r_{ui} = 1$ indicates an interaction between user $u$ and item $i$, and $s_{uv} = 1$ indicates a connection between users $u$ and $v$. Unobserved interactions or connections are denoted as 0.

A typical SR model uses encoder functions, like GCNs [7, 31], to learn user preference representations in both $\mathcal{G}_R$ and $\mathcal{G}_S$, which are then integrated to predict future user-item interactions. Formally, the SR backbone follows a unified framework:

$$\mathbf{z}_u = g(f_r(u, \mathcal{G}_R), f_s(u, \mathcal{G}_S)); \ \mathbf{z}_i = f_r(i, \mathcal{G}_R); \ \hat{r}_{ui} = Pred(\mathbf{z}_u, \mathbf{z}_i), \quad (1)$$

where $\mathbf{z}_u, \mathbf{z}_i \in \mathbb{R}^d$ represent the final embeddings of user $u$ and item $i$, respectively. The functions $f_r(.)$ and $f_s(.)$ denote the encoders for $\mathcal{G}_R$ and $\mathcal{G}_S$, while $g(.)$ combines the user representations from both graphs. $Pred(.)$ produces the prediction score, $\hat{r}_{ui}$, which represents the predicted preference score of user $u$ for item $i$. Then, the BPR loss [24] is used to optimize the SR task, which ensures that positive

interactions rank higher than negative ones:

$$\mathcal{L}_{bpr} = \sum_{(u,i^+,i^-)\in O^r} -\ln\sigma(\hat{r}_{ui^+} - \hat{r}_{ui^-}), \tag{2}$$

where $O^r = \{(u,i^+,i^-)|(u,i^+)\in\mathcal{R}^+, (u,i^-)\in\mathcal{R}^-\}$ is the training set; $\mathcal{R}^+$ is the observed positive sample set and $\mathcal{R}^-$ is the unobserved negative sample set; and $\sigma$ is the sigmoid function.

**Problem Definition.** Given $\mathcal{G}_R$, $\mathcal{G}_S$, and any SR backbone, our goal is to refine $\mathcal{G}_S$ by removing noisy edges and adding useful ones, thus enhancing SR's robustness against social noise and facilitate more accurate user-item interaction predictions in $\mathcal{G}_R$. This model-agnostic process is end-to-end with backbone training, allowing seamless integration into any existing SR backbone.

## 3.2 Diffusion Model (DM)

We first introduce DM based on DDPM [10], which is a foundation work widely used in the field of computer vision.

**Forward Process.** Given the input $s_0 \sim q(s_0)$, the forward process is a tractable Markov process that incrementally adds Gaussian noise over $T$ steps. Specifically, $s_t$ is derived by perturbing $s_{t-1}$, which can be formulated as:

$$q(s_t|s_{t-1}) = \mathcal{N}(s_t; \sqrt{1-\beta_t}s_{t-1}, \beta_t I), \tag{3}$$

where $\mathcal{N}$ is the Gaussian distribution; $\beta_t \in (0,1)$ controls the noise scale at time $t$. Based on the additivity property of independent Gaussian distributions, we can directly obtain $s_t$ from $s_0$:

$$q(s_t|s_0) = \mathcal{N}(s_t; \sqrt{\bar{\alpha}_t}s_0, (1-\bar{\alpha}_t)I), \tag{4}$$

where $\alpha_t = 1-\beta_t$ and $\bar{\alpha}_t = \prod_{t'=1}^{t}\alpha_{t'}$. Using the reparameterization trick, $s_t$ can be expressed as $\sqrt{\bar{\alpha}_t}s_0 + \sqrt{1-\bar{\alpha}_t}\epsilon$, with $\epsilon \sim \mathcal{N}(0,I)$. As $T \to \infty$, $s_T$ converges to standard Gaussian noise.

**Reverse Process.** The reverse process seeks to recover the original input $s_0$ from the corrupted $s_T$. While this process is modeled as a Markov chain, it is infeasible to derive its distribution due to the complexity of modeling high-dimensional distributions across time steps. Therefore, the posterior distribution is parameterized as:

$$p_\theta(s_{t-1}|s_t) = \mathcal{N}(s_t; \mu_\theta(s_t,t), \Sigma_\theta(s_t,t)), \tag{5}$$

where $\mu_\theta(s_t,t)$ and $\Sigma_\theta(s_t,t)$ are learnable predictors to approximate the Gaussian distribution. Nevertheless, when conditioned on $s_0$, the posterior becomes tractable and can be derived as:

$$q(s_{t-1}|s_t,s_0) = q(s_t|s_{t-1},s_0)\frac{q(s_{t-1}|s_0)}{q(s_t|s_0)} \propto \mathcal{N}(s_{t-1}; \tilde{\mu}(s_t,s_0,t), \tilde{\beta}_t I) \tag{6}$$

where $\tilde{\mu}(s_t,s_0,t)$ and $\tilde{\beta}_t$ are given by:

$$\tilde{\mu}(s_t,s_0,t) = \frac{\sqrt{\alpha_t}(1-\bar{\alpha}_{t-1})}{1-\bar{\alpha}_t}s_t + \frac{\sqrt{\bar{\alpha}_{t-1}}(1-\alpha_t)}{1-\bar{\alpha}_t}s_0, \quad \tilde{\beta}_t = \frac{(1-\bar{\alpha}_{t-1})(1-\alpha_t)}{1-\bar{\alpha}_t} \tag{7}$$

**Optimization of DM.** The parameters in DM are optimized by maximizing the evidence lower bound (ELBO) of the likelihood of $s_0$, which is equivalent to minimizing the KL-divergence between the posterior $q(s_{t-1}|s_t,s_0)$ in Eq. 6 and the learned approximate distribution $p_\theta(s_{t-1}|s_t)$ in Eq. 5 for $t \in \{1,\ldots,T\}$, commonly referred to as the denoising matching loss:

$$\mathcal{L}_t = \mathbb{E}_{q(s_t|s_0)}[D_{KL}(q(s_{t-1}|s_t,s_0) \| p_\theta(s_{t-1}|s_t))], \tag{8}$$

where $\mathcal{L}_t$ denotes the denoising matching loss at time $t$. In DDPM, the learning of $\Sigma_\theta(s_t,t)$ is omitted and set as $\beta_t I$ for training stability and simplification. As a result, $\mathcal{L}_t$ can be simplified as:

$$\mathcal{L}_t = \mathbb{E}_{q(s_t|s_0)}\left[\frac{1}{2\sigma^2(t)}\|\mu_\theta(s_t,t) - \tilde{\mu}(s_t,s_0,t)\|_2^2\right]. \tag{9}$$

Similar to Eq. 7, we can formulate $\mu_\theta(s_t,t)$ as:

$$\mu_\theta(s_t,t) = \frac{\sqrt{\alpha_t}(1-\bar{\alpha}_{t-1})}{1-\bar{\alpha}_t}s_t + \frac{\sqrt{\bar{\alpha}_{t-1}}(1-\alpha_t)}{1-\bar{\alpha}_t}\hat{s}_\theta(s_t,t), \tag{10}$$

where $\hat{s}_\theta(s_t,t)$ is the predictor of $s_0$, which is usually an MLP that takes $s_t$ and the time step embedding as the input. Substituting Eq. 7 and Eq. 10 into Eq. 9 yields the final loss:

$$\mathcal{L}_t = \mathbb{E}_{q(s_t|s_0)}\left[\frac{1}{2}\left(\frac{\bar{\alpha}_{t-1}}{1-\bar{\alpha}_{t-1}} - \frac{\bar{\alpha}_t}{1-\bar{\alpha}_t}\right)\|\hat{s}_\theta(s_t,t) - s_0\|_2^2\right]. \tag{11}$$

To summarize, DM is parameterized by the predictor $\hat{s}_\theta(s_t,t)$. During inference, the prediction is substituted into Eq. 10 to estimate the distribution, from which the next state is sampled. This process is repeated iteratively to perform the reverse denoising steps.

## 4 Proposed Method

**Model Overview.** We introduce ARD-SR, a model-agnostic social network refinement framework, which exploits DMs' powerful denoising and generative capability to eliminate noisy social connections and insert potentially valuable ones, to enhance the robustness of existing SR backbones. In particular, we propose personalized and robust social network generation with a position-specific noise schedule in the forward process based on user preference similarity and a reverse process guided by neighborhood preferences. Then, for an effective learning process, we devise a curriculum-based training mechanism to gradually introduce more difficult samples. Finally, ARD-SR is jointly trained with the SR backbone, allowing for progressive mutual enhancement of the social network and the learned user representations, ultimately resulting in a more robust SR. The overall framework is depicted in Figure 1.

### 4.1 Forward and Reverse Process of ARD-SR

We perform the forward and reverse processes in the graph space. The forward process takes each row of the social adjacency matrix $S$ as the input at $t = 0$, denoted as $s_0^u$, which corresponds to the connection of user $u$ to other users in the original social network:

$$s_0^u = [s_{u1},\ldots,s_{uv},\ldots,s_{um}], \tag{12}$$

where $s_{uv}$ is the ground truth value. Unlike the image domain, each entry in the input represents a meaningful relationship between a specific pair of users. Moreover, the inputs are sparse binary vectors and there exists stronger long-range dependencies among users, making it challenging for the DM to learn the noise patterns. To account for this, we tailored both the forward and reverse processes to enable personalized and robust social network refinement.

#### 4.1.1 *Forward Process with Position-specific Noise Schedule*. Starting from the initial state $s_0^u$ of user $u$, the forward process unfolds by incrementally adding Gaussian noise to $s_0^u$ over $T$ steps, as defined in Eq. 3. The variance $\beta_t$ is typically scheduled using a linear approach, defined as $\eta\left[\beta_{\min} + \frac{t-1}{T-1}(\beta_{\max} - \beta_{\min})\right]$, where $\eta$ is the noise scale; $\beta_{\min}$ and $\beta_{\max}$ are the lower and upper bound of the noise. Note that $\beta_t$ is a scalar applied uniformly across $s_{t-1}^u$. However, unlike the image domain where the inputs lack semantic meaning, each element in $s_{t-1}^u$ corresponds to the connection between user $u$ and every other single user.

To better simulate how social networks respond to noise perturbations, we introduce an adaptive noise schedule informed by user preferences. Previous study [28] has shown that social connections

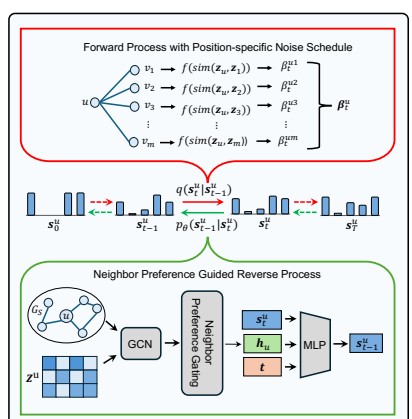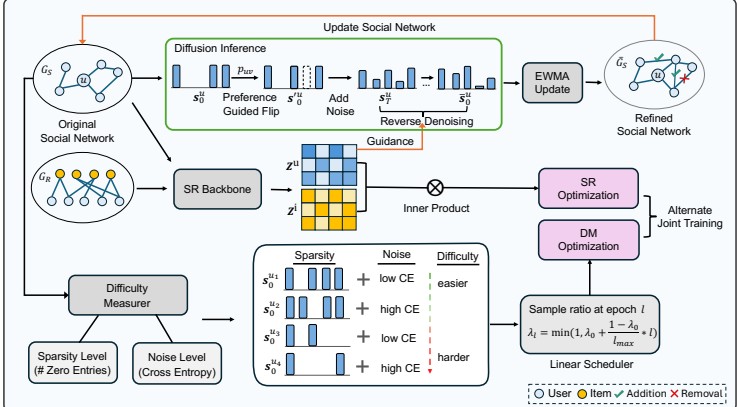

**Figure 1: The framework of ARD-SR. The left part is the forward and reverse process (Section 4.1). The right shows the ARD-SR's curriculum-based training (Section 4.2) and its mutual enhancement with the SR backbone via joint training (Section 4.3).**

with higher preference similarity contribute more to preference modeling. Consequently, connections between users $u$ and $v$ with higher preference similarity are less likely to be noisy and should therefore undergo smaller perturbation during the forward process. Thus, we scale the global noise variance $\beta_t$ with position-specific factors. For a user pair $(u, v)$, it is defined as:

$$\beta_t^{uv} = \beta_t \cdot \gamma_{uv}, \quad \gamma_{uv} = f(sim\langle \mathbf{z}_u, \mathbf{z}_v \rangle), \tag{13}$$

where $\gamma_{uv}$ represents the adaptive scale for the corresponding position, which is a function of the cosine similarity between the embeddings $\mathbf{z}_u$ and $\mathbf{z}_v$, derived from the SR backbone as in Eq. 1. We empirically choose $f(x) = 1 - w \cdot exp(kx)$, where $w$ and $k$ are the hyperparameters. This ensures (1) a monotone decreasing function such that users with higher preference similarity are subjected to smaller noise perturbations; and (2) the scaling factor stays below 1 and approaches 1 asymptotically, ensuring stability in the forward process and converging smoothly to a standard Gaussian distribution. The forward process is then reformulated as:

$$q(\mathbf{s}_t^u | \mathbf{s}_{t-1}^u) = N(\mathbf{s}_t^u; \sqrt{\mathbf{1} - \boldsymbol{\beta}_t^u} \odot \mathbf{s}_{t-1}^u, diag(\boldsymbol{\beta}_t^u)), \tag{14}$$

where $\boldsymbol{\beta}_t^u = (\beta_t^{u1}, \dots, \beta_t^{um})$; $\odot$ is the vector element-wise product and $diag$ represents the diagonal matrix. Similar to Eq. 4, $\mathbf{s}_t^u$ is directly obtainable from the input $\mathbf{s}_0^u$, where the $v$-th element $s_t^{uv}$ is reformulated as $s_t^{uv} = \sqrt{\bar{\alpha}_t^{uv}} s_{uv} + \sqrt{1 - \bar{\alpha}_t^{uv}} \epsilon$, with $\epsilon \sim \mathcal{N}(0, 1)$.

*4.1.2 **Neighbor Preference Guided Reverse Process.*** The reverse process learns to gradually restore $\mathbf{s}_0^u$ from the corrupted sample. As derived in Eq. 10, it is parameterized by $\hat{\mathbf{s}}_\theta(\mathbf{s}_t^u, t)$, which predicts $\mathbf{s}_0^u$ based on $\mathbf{s}_t^u$ and $t$. However, relying solely on $\mathbf{s}_t^u$ does not guarantee high-quality recovery due to its high sparsity and insufficient guidance for capturing long-range dependencies among users. Moreover, it is also impractical to perform the reverse process for users with no observed connections.

To address this, we condition the predictor on each user's neighborhood preference to guide the reverse process toward personalized generation. Specifically, we design a gating mechanism based on the user preference representation and those of their neighbors, which filters the latent aspects of the users' preferences that are relevant for driving connections with others. The filtered user

representation is denoted as:

$$\mathbf{h}_u = \mathbf{z}_u \odot \sigma \left( \mathbf{W} \left( \mathbf{z}_u \odot \text{GCN} \left( sg[\![\mathbf{Z}^u]\!], \mathcal{G}_s \right)_u \right) + \mathbf{b} \right), \tag{15}$$

where $\mathbf{Z}^u \in \mathbb{R}^{m \times d}$ denotes the final user embedding matrix from the SR backbone; $\text{GCN}(.)_u$ is a lightweight GCN similar to Light-GCN [8], capturing both immediate and multi-hop neighbors' preferences, thereby effectively modeling long-range dependencies. The stop-gradient operator $sg[\![.]\!]$ prevents gradients of ARD-SR from affecting the backbone's parameters, ensuring stable updates for the backbone. The predictor is thus reformulated as:

$$\hat{\mathbf{s}}_\theta(\mathbf{s}_t^u, t, \mathbf{h}_u) = MLP(\mathbf{s}_t^u, t, \mathbf{h}_u). \tag{16}$$

Conditioning the predictor on $\mathbf{h}_u$ guides the reverse process to align with each user's distinct preference for building connections, facilitating personalized social network generation.

## 4.2 Optimization of ARD-SR

**Objective Function.** The optimization of the predictor structure, $\hat{\mathbf{s}}_\theta(\mathbf{s}_t^u, t, \mathbf{h}_u)$, follows a similar approach to Eq. 11. However, the adapted noise schedule introduces position-specific weights in the loss function, which is reformulated as:

$$\mathcal{L}_t = \mathbb{E}_{q(\mathbf{s}_t^u | \mathbf{s}_0^u)} \left[ \frac{1}{2} \sum_{i=1}^m \left( \frac{\tilde{\alpha}_{t-1}^{uv}}{1 - \tilde{\alpha}_{t-1}^{uv}} - \frac{\tilde{\alpha}_t^{uv}}{1 - \tilde{\alpha}_t^{uv}} \right) \left( [\hat{\mathbf{s}}_\theta(\mathbf{s}_t^u, t, \mathbf{h}_u)]_v - s_{uv} \right)^2 \right], \tag{17}$$

where $\tilde{\alpha}_{t-1}^{uv} = \prod_{t'=1}^t \alpha_{t'}^{uv}$ and $\alpha_t^{uv} = 1 - \beta_t^{uv}$.

**Progressive Training with Curriculum Learning.** Some rows of the social adjacency matrix exhibit high sparsity, making it difficult for the DM to capture meaningful patterns. Moreover, even rows with low sparsity can still pose challenges when they contain significant noise. As such, we design a curriculum-based training mechanism that starts training with simple samples and gradually introduces more difficult ones. This allows the DM to strengthen its denoising ability as training advances progressively.

Specifically, we use sparsity and noise level as the difficulty measurer of input samples, considering rows with both low sparsity and low noise levels to be easier samples. Sparsity is measured by the number of zero entries in each row of $\mathbf{S}$:

$$\text{Sparsity}(u) = \sum_{v=1}^m \mathbb{I}_{\{s_{uv}=0\}}, \tag{18}$$

where $\text{Sparsity}(u)$ denotes the sparsity level of user $u$ and $\mathbb{I}$ is the indicator function. For noise level, we use binary cross-entropy

---

**Algorithm 1:** The overall process of ARD-SR

---

**Input:** $\mathcal{G}_S$, $\mathcal{G}_R$, $\mathbf{S}$, $\mathbf{R}$, SR backbone
**Output:** Refined social network and optimal model parameters

1  Initialize backbone parameter $\Theta_1$ and ARD-SR parameters $\Theta_2$;
2  **for** $l = 1$ to $max\_epoch$ **do**
3    Train SR backbone on $\mathcal{G}_S$ and $\mathcal{G}_R$ to optimize $\Theta_1$ w.r.t Eq. 2;
4    **if** $l > 10$ **then** // Start Joint Training
5      Calculate $h_u$ via Eq. 15;
6      Calculate $\lambda_l$ by Eq. 21;
7      M ← the number of users having at least one social neighbor;
8      **foreach** $u \in \mathcal{U}$ **do** // Curriculum-based DM Training
9        Calculate Difficulty($u$) by Eq. 20 ;
10       **if** $rank(\text{Difficulty}(u)) \leq \lambda_l * M$ **then**
11         Sample $t \sim \mathcal{U}(1, T)$;
12         Compute $\hat{s}_\theta(\mathbf{s}_t^u, t, \mathbf{h}_u)$ via Eq. 16 ;
13         Take gradient descent on $\mathcal{L}_t$ in Eq. 17 to optimize $\Theta_2$;

14     **if** $l \% 5 == 0$ **then** // Social Network Refinement
15       **foreach** $u \in \mathcal{U}$ **do**
16         **for** $v = 1, \ldots, m$ **do**
17           Flip $s_{uv}$ based on $p_{uv}$ in Eq. 22;
18         Calculate $\mathbf{s}_T^u$ given $\mathbf{s}_0^u$ based on Eq. 14;
19         **for** $t = T, \ldots, 1$ **do** // Reverse Denoising
20           Compute $\mathbf{s}_{t-1}^u$ with $\hat{s}_\theta(\mathbf{s}_t^u, t, \mathbf{h}_u)$ via Eq. 23;
21       Update EWMA matrix $\bar{\mathbf{S}}_l$ via Eq. 24;
22       Update $\mathbf{S}$ and $\mathcal{G}_S$ based on $\bar{\mathbf{S}}_l$;

---

loss, which reflects uncertainty in the model's predictions. A higher loss typically indicates a higher noise level, as noted in previous work [9, 34]. The noise level of user $u$ is calculated as:

$$\text{Noise}(u) = -\sum_{v=1}^{m} \mathbb{I}_{\{s_{uv}=1\}} \left( s_{uv} \log(\bar{s}_{uv}) + (1 - s_{uv}) \log(1 - \bar{s}_{uv}) \right) \quad (19)$$

where $\bar{s}_{uv}$ is the exponential weighted moving average of the predicted connection score, which will be elaborated in Eq. 24. We selectively accumulate the cross-entropy loss over non-zero entries only, thereby reducing the influence of the abundant zero entries and emphasizing the informative non-zero entries for a more accurate noise assessment. The final difficulty of each sample is measured by combining the two metrics using a simple rank aggregation method, given by,

$$\text{Difficulty}(u) = \text{rank}(\text{Sparsity}(u)) + \text{rank}(\text{Noise}(u)). \quad (20)$$

Accordingly, we thus use the linear scheduler to gradually introduce the training samples to the diffusion training, parameterized by $\lambda_l$:

$$\lambda_l = \min\left(1, \lambda_0 + \frac{1 - \lambda_0}{l_{\max}} \cdot l\right), \quad (21)$$

where $\lambda_l$ denotes the proportion of easiest examples used for training in epoch $l$; $l_{max}$ is the epoch when $\lambda_l$ reaches 1, after which all samples are used for training.

## 4.3 Iterative Social Network Refinement

We now present how ARD-SR integrates with the SR backbone to iteratively refine the social network throughout the training process, thus enhancing the robustness of existing SR backbones. The overall process is described in Algorithm 1.

### 4.3.1 *Preference Guided Flip for Enhanced Input.* We employ ARD-SR's reverse process to generate a row vector for each user, which guides the social network refinement. Image generation tasks typically start from a random standard Gaussian sample, which

compromises the personalized social network generation. A simple solution [35] to retain personalized information is to add Gaussian noise to the original row vector $\mathbf{s}_0^u$ and then denoise it. However, the social adjacency matrix is highly sparse. When Gaussian noise is added, the few non-zero entries can be easily overwhelmed by the noise applied to the zero entries, obscuring personalized information. Additionally, some zero entries may represent false negatives, making it challenging for the model to accurately distinguish them from true zeros and correctly infer hidden connections.

Hence, we apply a random flip operation to the original row vector based on user preference similarity. The probability of flipping a user pair $(u, v)$ is defined as:

$$p_{uv} = \begin{cases} sigmoid(-\text{sim}\langle \mathbf{z}_u, \mathbf{z}_v \rangle) & \text{if } s_{uv} = 1 \\ sigmoid(\text{sim}\langle \mathbf{z}_u, \mathbf{z}_v \rangle) & \text{if } s_{uv} = 0 \end{cases} \quad (22)$$

This prioritizes the flipping of low-similarity existing edges or high-similarity unobserved edges, enhancing the initial input quality and mitigating the overwhelming effect of zero entries, which leads to a more efficient and personalized generation process. The flipped vectors are then corrupted with Gaussian noise via Eq. 14, yielding $\mathbf{s}_T^u$, which is passed to the iterative reverse denoising for $T$ steps. Following previous work [35], we ignore the variance and let $\mathbf{s}_{t-1}^u = \boldsymbol{\mu}_\theta(\mathbf{s}_t^u, t, \mathbf{h}_u)$ for deterministic inference, where $\boldsymbol{\mu}_\theta(\mathbf{s}_t^u, t, \mathbf{h}_u)$ is the predicted mean of the posterior distribution $p_\theta(\mathbf{s}_{t-1}^u | \mathbf{s}_t^u)$. Similar to Eq. 10, it is reformulated as:

$$\left[\boldsymbol{\mu}_\theta(\mathbf{s}_t^u, t, \mathbf{h}_u)\right]_v = \frac{\sqrt{\alpha_t^{uv}}(1 - \bar{\alpha}_{t-1}^{uv})}{1 - \bar{\alpha}_t^{uv}} s_t^{uv} + \frac{\sqrt{\bar{\alpha}_{t-1}^{uv}}(1 - \alpha_t^{uv})}{1 - \bar{\alpha}_t^{uv}} \left[\hat{s}_\theta(\mathbf{s}_t^u, t, \mathbf{h}_u)\right]_v. \quad (23)$$

### 4.3.2 *Progressive Mutual Enhancement with SR Backbone.* Following [13], we first warm up the SR backbone by training it alone for 10 epochs to ensure reliable user representations that guide both the position-specific noise schedule and the reverse process. After the warm-up, we alternate training between ARD-SR and the SR backbone in each epoch for progressive mutual enhancement. The backbone's learned user representations inform the position-specific noise schedule and enhance the ARD-SR's reverse process, while ARD-SR refines the social network, which in turn helps the backbone learn more robust user representations. This iterative enhancement allows both components to progressively improve. The social network is refined every 5 epochs based on the ARD-SR inference prediction and then used for further training of both the backbone and ARD-SR.

To smooth fluctuation and ensure consistency of the iterative social network refinement, we exploit an exponential weighted moving average (EWMA) [12] of the reverse process predictions, which accounts for both current and preceding steps:

$$\bar{\mathbf{S}}_l = \tau \hat{\mathbf{S}}_l + (1 - \tau) \bar{\mathbf{S}}_{l-1}, \quad (24)$$

where $\bar{\mathbf{S}}_l$ denotes the EMA of the predicted social adjacency matrix at epoch $l$; $\hat{\mathbf{S}}_l$ is the predicted matrix at epoch $l$; $\tau$ is the decay factor. $\bar{\mathbf{S}}_l$ is then used to refine the social network by removing low-scoring edges and inserting high-scoring ones. Specifically, edges with EMA scores below a predefined threshold $\rho$ are considered noise and removed. To maintain stability, each iteration deletes no more than 1% of the total edges, prioritizing those with the lowest scores. $\rho$ decays over time, and if the number of removable edges exceeds the 1% limit, the threshold is scaled by a decay factor.

This enables smoother convergence of the refinement process by gradually decreasing the strictness of edge removal as the overall network quality improves. After deletion, an equal number of unobserved social edges with the highest prediction scores are inserted, including those previously removed, to facilitate convergence.

## 4.4 Complexity Analysis

ARD-SR is a model-agnostic framework, and its additional complexity over the SR backbone comes from the DM, both in training and inference. For training, the primary complexity is the 2-layer MLP predictor, while for inference, it arises from the iterative reverse denoising process. The complexity of the MLP is $O((m + d + d_t) \times d_h + m \times d_h)$, where $d_t$ and $d_h$ represent dimensions of the time step embedding and hidden layer, respectively. Since $d, d_t, d_h \ll m$, the overall complexity is dominated by $O(m \times d_h)$, making it linearly scalable with the number of users $m$. One drawback of the DM is its slow inference, especially for large $T$, due to the iterative nature of the Markov chain in the reverse denoising process. We address it using the DDIM approach [27], which accelerates inference by relaxing the Markov chain assumption and reparameterizing the reverse process. Specifically, it reduces the original $T$ to a fraction of $T$, i.e., $t = \{1, 2, \ldots, T/\zeta\}$, accelerating inference by a factor of $\zeta$ without compromising performance.

## 5 Experiments

### 5.1 Experiment Settings

*5.1.1 Datasets.* We adopt three commonly used real-world datastes, i.e., Ciao [30], Douban[1] and FilmTrust [5], for evaluation. All datasets are based on explicit ratings, and, following prior work [8, 36], we remove ratings below 3 for Ciao and Douban, and below 2 for FilmTrust. Users and items with less than two interactions are removed. User-item interactions are sorted chronologically and split into training, validation, and testing sets with an 8:1:1 ratio. Detailed statistics are provided in Table 1.

*5.1.2 Baselines.* We compare ARD-SR with five robust SR approaches, including (1) *social network refinement methods*: Rule-based, GDMSR [23], SHaRe [13] and GBSR [41]; and (2) the *DM-based method* RecDiff [19]. We do not compare with GDSSL [18] due to the absence of available code and insufficient model details in the original paper. In particular, **Rule-based** removes social connections with few common co-consumed items in the user-item interaction networks; **GDMSR** trains a link prediction model to remove noisy edges; **SHaRe** iteratively refines the social network based on the user embedding similarity; **GBSR** utilizes self-supervised contrastive learning with dropout-based graph augmentation to learn the noise pattern; **RecDiff** trains a DM to denoise users' social space embedding. For each selected method, we choose three representative state-of-the-art SR models as the backbone, including the traditional matrix factorization based TrustSVD [6], and the GNN-based DiffNet++ [37] and MHCN [46].

*5.1.3 Evaluation.* We utilize three commonly used metrics: HR@K, NDCG@K, and MRR@K to assess the performance of all methods.

---

[1]https://pan.baidu.com/s/1hrJP6rq

**Table 1: Statistics of experiment datasets.**

|  | Ciao | Douban | FilmTrust |
|---|---|---|---|
| # Users | 7,291 | 2,668 | 1,336 |
| # Items | 17,876 | 15,940 | 793 |
| # Interactions | 140,628 | 535,210 | 33,363 |
| # Social Relations | 57,544 | 32,705 | 1,484 |
| Interaction Density | 0.044% | 0.309% | 1.473% |
| Social Relation Density | 0.108% | 0.460% | 0.083% |

In accordance with [8, 36], we rank all non-interacted items for each user, ensuring an unbiased evaluation.

*5.1.4 Implementation Details.* All SR models are trained to optimize the BPR loss. We use Adam [17] as the optimizer, with a fixed batch size of 1024, a learning rate of 0.001, and a latent embedding dimension of 64. For a fair comparison, we first identify the optimal parameter configuration for each SR backbone and keep them fixed. On this basis, we integrate the selected robust SR approaches and tune their hyperparameters. The key hyperparameters of SR backbones and robust SR approaches are tuned via extensive grid search based on the validation set performance. We adopt the early stop strategy to terminate training if the performance of the backbone on the validation set does not improve for 50 consecutive epochs. For ARD-SR[2], we train the DM part with an independent optimizer separate from the SR backbone, with a fixed batch size of 64 and learning rate searched over $\{e^{-2}, e^{-3}, e^{-4}\}$. The dimension of the time step embedding is fixed at 16. We empirically set $\lambda_0$ to 0.4, $l_{\max}$ to 50, $\zeta$ to 10 and $\rho$ to 0.6; and the number of layers in the GCN encoder is set to 3. For other essential parameters, we search the total diffusion step $T$ over $\{10, 30, 50, 100, 200\}$; the noise scale $\eta$ is searched in $\{e^{-1}, e^{-2}, e^{-3}, e^{-4}\}$; the noise bound $\beta_{\min}$ and $\beta_{\max}$ are searched in $\{e^{-2}, e^{-3}, e^{-4}\}$ and $\{e^{-1}, e^{-2}, e^{-3}\}$, respectively; and $\tau$ is searched in $\{0.1, 0.3, 0.5, 1\}$.

## 5.2 Comparative Results and Analysis

*5.2.1 Overall Performance.* Table 2 presents the performance of three backbones integrated with six robust SR methods across three datasets, where "base" represents the vanilla backbone. Each integration was evaluated five times, and the average results are provided to ensure reliability. We report the relative improvement of ARD-SR (bolded) over the runner-up baselines (underlined) and perform t-tests, yielding statistically significant results with a $p$-value $< 0.001$. Several major findings are noted. **Firstly**, all backbones exhibit improvement after integrating robust SR methods in most cases, implying the importance of enhancing social signals for robust SRs. **Secondly**, among the social network refinement methods, the model-based GDMSR and GBSR outperform the heuristic-based (i.e. Rule-based and SHaRe) and the performance even drops in certain cases when Rule-based is applied, e.g., the performance on Ciao with Rule-based + MHCN. This indicates that simple heuristics do not necessarily correlate with effective refinement. **Thirdly**, compared with RecDiff that denoise the latent social space with DM, ARD-SR has better performance. This suggests the effectiveness of directly using DM to refine the social network. **Lastly**, ARD-SR demonstrates superior performance, with an average improvements of 4.81% in HR, 6.95% in NDCG, and 6.80% in MRR across all datasets. This is mainly attributed to its three key designs:

---

[2]Our code is available at https://anonymous.4open.science/r/ARD-SR-4C34.

**Table 2: Overall performance of ARD-SR on three datasets ($p - \text{value} < 0.001$).**

| SR Backbone | Robust SR | Ciao | | | Douban | | | FilmTrust | | |
|---|---|---|---|---|---|---|---|---|---|---|
| | | HR@10 | NDCG@10 | MRR@10 | HR@10 | NDCG@10 | MRR@10 | HR@10 | NDCG@10 | MRR@10 |
| TrustSVD | base | 0.0771 | 0.0271 | 0.0279 | 0.4571 | 0.0947 | 0.2045 | 0.6175 | 0.3051 | 0.2694 |
| | Rule-based | 0.0778 | 0.0279 | 0.0283 | 0.4727 | 0.0943 | 0.2053 | 0.6211 | 0.3094 | 0.2731 |
| | GDMSR | 0.0806 | 0.0301 | 0.0305 | 0.4882 | 0.0985 | 0.2145 | 0.6441 | 0.3285 | 0.2834 |
| | SHaRe | 0.0787 | 0.0285 | 0.0288 | 0.4812 | 0.0951 | 0.2087 | 0.6310 | 0.3197 | 0.2810 |
| | GBSR | 0.0802 | 0.0299 | 0.0293 | 0.5026 | 0.1002 | 0.2133 | 0.6412 | 0.3228 | 0.2823 |
| | RecDiff | 0.0794 | 0.0291 | 0.0296 | 0.4927 | 0.0977 | 0.2099 | 0.6356 | 0.3243 | 0.2809 |
| | **ARD-SR** | **0.0845** | **0.0316** | **0.0323** | **0.5411** | **0.1125** | **0.2326** | **0.6633** | **0.3440** | **0.2956** |
| | Improve | 4.84% | 4.98% | 5.90% | 7.66% | 12.27% | 8.44% | 2.98% | 4.72% | 4.30% |
| DiffNet++ | base | 0.0570 | 0.0186 | 0.0191 | 0.4087 | 0.0754 | 0.1626 | 0.6305 | 0.3179 | 0.2787 |
| | Rule-based | 0.0583 | 0.0190 | 0.0195 | 0.4110 | 0.0761 | 0.1641 | 0.6244 | 0.3118 | 0.2719 |
| | GDMSR | 0.0610 | 0.0205 | 0.0211 | 0.4421 | 0.0802 | 0.1712 | 0.6421 | 0.3329 | 0.2847 |
| | SHaRe | 0.0591 | 0.0194 | 0.0198 | 0.4367 | 0.0791 | 0.1701 | 0.6310 | 0.3110 | 0.2755 |
| | GBSR | 0.0603 | 0.0197 | 0.0207 | 0.4402 | 0.0795 | 0.1709 | 0.6447 | 0.3343 | 0.2819 |
| | RecDiff | 0.0615 | 0.0210 | 0.0203 | 0.4563 | 0.0813 | 0.1741 | 0.6512 | 0.3337 | 0.2839 |
| | **ARD-SR** | **0.0637** | **0.0217** | **0.0223** | **0.4793** | **0.0910** | **0.1881** | **0.6778** | **0.3492** | **0.3013** |
| | Improve | 3.58% | 3.33% | 5.69% | 5.04% | 11.93% | 8.04% | 4.08% | 4.46% | 5.83% |
| MHCN | base | 0.0797 | 0.0287 | 0.0297 | 0.4736 | 0.1054 | 0.2243 | 0.6628 | 0.3306 | 0.2900 |
| | Rule-based | 0.0793 | 0.0276 | 0.0283 | 0.4817 | 0.1078 | 0.2257 | 0.6691 | 0.3302 | 0.2875 |
| | GDMSR | 0.0825 | 0.0296 | 0.0312 | 0.4910 | 0.1126 | 0.2321 | 0.6811 | 0.3391 | 0.2968 |
| | SHaRe | 0.0810 | 0.0289 | 0.0303 | 0.4858 | 0.1101 | 0.2298 | 0.6713 | 0.3331 | 0.2895 |
| | GBSR | 0.0817 | 0.0305 | 0.0309 | 0.4878 | 0.1109 | 0.2279 | 0.6623 | 0.3401 | 0.3031 |
| | RecDiff | 0.0815 | 0.0299 | 0.0310 | 0.4840 | 0.1115 | 0.2307 | 0.6698 | 0.3387 | 0.2947 |
| | **ARD-SR** | **0.0879** | **0.0330** | **0.0336** | **0.5140** | **0.1187** | **0.2532** | **0.7073** | **0.3636** | **0.3219** |
| | Improve | 6.55% | 8.20% | 7.69% | 4.68% | 5.42% | 9.09% | 3.85% | 7.23% | 6.20% |

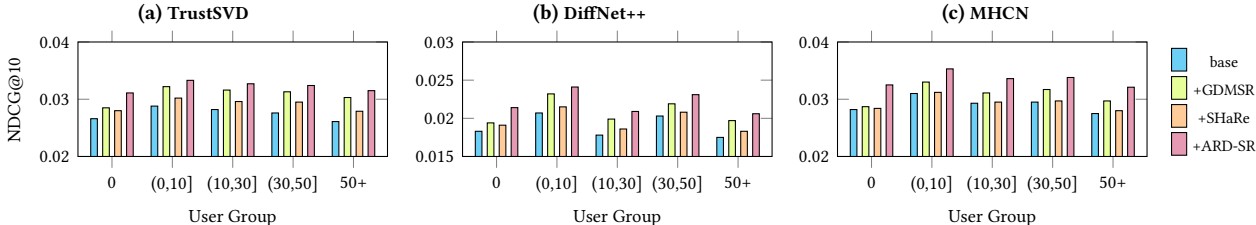

**Figure 2: Performance comparison across user groups with different numbers of social neighbors w.r.t NDCG@10.**

(a) the position-specific noise schedule in the forward process and the neighborhood preference-guided reverse process, which facilitates personalized and robust social network generation; (b) the curriculum-based training mechanism, resulting in enhanced denoising capabilities as training progresses; and (c) the iterative social network refinement, strengthened by preference-guided flip operations and mutual enhancement with the SR backbone.

### 5.2.2 *Performance w.r.t Different Social Sparsity Levels.* The social network exhibits significant user disparities, presenting challenges for effective refinement. For example, some users have very few or even no neighbors; others may have many neighbors but a significant portion of their connections could be noisy. To assess whether the social network refinement by ARD-SR benefits all users, we divide users into five groups based on their number of neighbors in the original social network. The performance of three SR backbones on Ciao, integrated with different robust SR methods, is reported in Figure 2. First, we observe that having more social neighbors does not necessarily result in better performance. For example, users in the "50+" group exhibit the poorest results, likely due to the increased noise in their connections, which impairs the model's ability to infer users' preferences based on the social neighborhood. This aligns with our training curriculum design, which treats users with both low sparsity and low noise levels as easier

samples. Moreover, GDMSR has a limited impact on users with no neighbors in contrast to its stronger impact on other groups, as its refinement mechanism only considers removing noisy edges. In contrast, the backbone integrated with ARD-SR consistently demonstrates superiority across all groups, including the group with no neighbors. This highlights the effectiveness of ARD-SR in generating high-quality social networks for more robust SR.

### 5.2.3 *A Statistical Analysis of the Refined Social Network.* We further perform a statistical analysis to investigate how the refined social network changes with the integration of ARD-SR. The result of integrating ARD-SR into MHCN is given in Table 3, where preference similarity is measured by the cosine similarity between the final user representations obtained from the MHCN backbone. Firstly, the proportion of noisy edges removed is 27.1%, 18.3%, and 24.7% for the three datasets, respectively. An equal number of potentially useful edges are added, resulting in a substantial increase in the proportion of users with at least one social neighbor, especially notable on Ciao (29.6% → 100%) and FilmTrust (34.2% → 100%). Such transformation not only eliminates noise from existing user connections but also empowers SR backbones to uncover new potential social influence for users who previously lack neighbors, thereby enhancing the robustness of SR. Secondly, the refined social

**Table 3: Comparison of the social network statistics with the integration of ARD-SR.**

| Dataset | Refinement Stage | Edge Removal (%) | Users with Neighbors (%) | Average Preference Similarity |
|---------|------------------|------------------|--------------------------|-------------------------------|
| Ciao | Before | - | 29.6 | 0.454 |
| | After | 27.1 | 100 ↑ | 0.513 ↑ |
| Douban | Before | - | 96.6 | 0.442 |
| | After | 18.3 | 100 ↑ | 0.485 ↑ |
| FilmTrust | Before | - | 34.2 | 0.694 |
| | After | 24.7 | 100 ↑ | 0.737 ↑ |

network exhibits an increase in the average user preference similarity across all datasets, which is confirmed by the independent t-tests with all $p$-values < 0.001. This well supports the design of our position-specific noise schedule, the neighbor preference-guided reverse process, and the preference-guided flip operation, whose effectiveness will be verified in section 5.3.

## 5.3 Ablation Study

We conduct ablation studies to assess the contributions of key components in ARD-SR. The results, which use MHCN as SR backbone, are presented in Table 4, with several critical insights noted.

*Effect of Position-specific Noise Schedule.* In the forward process of ARD-SR, we introduce a position-specific noise schedule to better simulate how the social network reacts to noise corruption. To assess its impact, we replace it with a conventional global linear schedule (denoted as "$w/o$ AS"). The performance reduction in "$w/o$ AS" underscores its effectiveness in facilitating more accurate learning of the social network's underlying structure, thereby improving the recommendation performance.

*Effect of Neighborhood Preference Guidance.* In the reverse process, we incorporate each user's neighborhood preference into the predictor to guide the denoising process, facilitating a personalized generation. We exclude this feature from the predictor in Eq. 16 (shortened as "$w/o$ guide") and observe a decline in the recommendation accuracy, showcasing the importance of neighbor preference in guiding the reverse denoising.

*Effect of Curriculum Learning.* Curriculum learning is integrated into ARD-SR training to gradually introduce difficult samples as its denoising capability improves. We assess its effectiveness by removing this mechanism (denoted as "$w/o$ CL"), leading to diminished recommendation accuracy. This confirms that curriculum learning helps ensure a smooth and effective learning process.

*Effect of Preference Guided Flip.* It introduces preference-guided corrections to the input for the reverse denoising process, effectively addressing the sparsity issue and adding useful signals. To examine its effect, we compare it with its variant, which directly operates on the original vector (shortened as "$w/o$ flip"). Its performance is worse than ARD-SR, indicating the effectiveness of the flipping operation in enhancing the refinement quality.

## 5.4 Parameter Sensitivity Analysis

We now examine the influence of key hyperparameters on ARD-SR's performance, focusing on the total diffusion steps $T$, noise scale $\eta$, and the EMA decay coefficient $\tau$. The trends, with MHCN as the backbone, are presented in Figure 3, showing similar patterns across all datasets for each selected hyperparameter. Generally,

**Table 4: Ablation studies of key components in ARD-SR.**

| Dataset | Ciao | | | Douban | | | FilmTrust | | |
|---------|------|------|------|--------|------|------|-----------|------|------|
| Metric | H@10 | N@10 | M@10 | H@10 | N@10 | M@10 | H@10 | N@10 | M@10 |
| ARD-SR | **0.0879** | **0.0330** | **0.0336** | **0.5140** | **0.1187** | **0.2532** | **0.7073** | **0.3636** | **0.3219** |
| $w/o$ AS | 0.0842 | 0.0314 | 0.0319 | 0.5002 | 0.1151 | 0.2497 | 0.6931 | 0.3584 | 0.3153 |
| $w/o$ guide | 0.0850 | 0.0319 | 0.0324 | 0.4987 | 0.1155 | 0.2505 | 0.6943 | 0.3601 | 0.3167 |
| $w/o$ flip | 0.0867 | 0.0326 | 0.0331 | 0.5094 | 0.1170 | 0.2520 | 0.7043 | 0.3628 | 0.3204 |
| $w/o$ CL | 0.0861 | 0.0322 | 0.0328 | 0.5053 | 0.1164 | 0.2512 | 0.7029 | 0.3611 | 0.3182 |

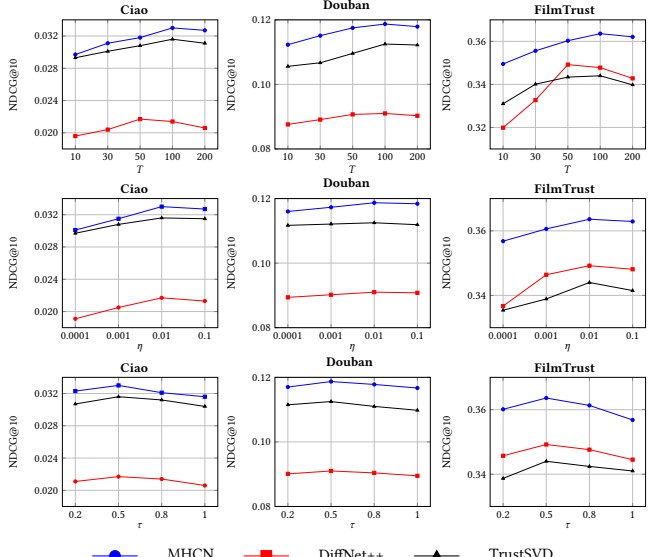

**Figure 3: Impacts of ARD-SR's key hyper-parameters.**

the performance increases as the values of these hyperparameters rise, until a peak is reached, after which the performance starts to decline. Specifically, we observe that the optimal value of $\eta$ is around 0.01, while the ideal $T$ is either 50 or 100. For $\tau$, values near 0.5 consistently yield the best results across different datasets. These findings highlight the importance of carefully tuning ARD-SR's hyperparameters to achieve effective social network refinement and maximize recommendation accuracy.

## 6 Conclusion

In this work, we propose ARD-SR, a novel model-agnostic diffusion-based social network refinement framework for robust social recommendation. For more effective social network generation with the diffusion model, we introduce a position-specific noise schedule into the forward process, effectively simulating how social networks respond to noise perturbations. Meanwhile, the reverse process is conditioned on each user's neighborhood preferences, enabling personalized network refinement. To further improve learning efficacy, we propose a curriculum-based training mechanism to progressively introduce more challenging samples as its denoising capability strengthens. Finally, by alternating the training of ARD-SR and the SR backbone iteratively, we ensure continuous enhancement of both the social network and user preference modeling. Experiments across three real-world datasets demonstrate that ARD-SR consistently enhances the performance of SR backbones, providing improved robustness and recommendation accuracy.

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

Received 20 February 2007; revised 12 March 2009; accepted 5 June 2009

