# OpenReview forum: "Model-Agnostic Social Network Refinement with Diffusion Models for Robust Social Recommendation"
_ACM.org/TheWebConf/2025/Conference — WWW 2025 Oral_

### Official Review · Reviewer_ju8x · 2024-11-30

**Novelty:** 5
**Technical Quality:** 5

**Review:**

This manuscript proposes a model-agnostic framework that integrates social network information into recommendation systems through DM. It identifies and addresses challenges faced when applying DM to the recommendation domain. Additionally, the manuscript provides a detailed analysis of the algorithm's time complexity, conducts experiments on datasets of varying scales, and validates the effectiveness of the proposed modules.

(1) To address the issue of inherent noise in social networks affecting recommendation performance, the manuscript introduces an adaptive module driven by user preferences. Experimental results demonstrate its contribution to alleviating this problem.

(2) The proposed framework is model-agnostic, capable of adapting to various recommendation algorithms without requiring significant modifications to the model structure.

**Questions:**

(1) The introduction should highlight the distinction between this work and the recent social recommendation approach, RecDiff, in order to emphasize its novelty.

(2) In the experiment, what are the encoders fr and fs used to obtain the final embeddings zu and zi for user u and item i? This is crucial for replicating the experimental results.

(3) What does the rank in Equation (20) specifically refer to? It is recommended to clarify this in the original text.

(4) In the subfigure on the right side of Figure 1, it would be helpful to indicate the execution order of the algorithm steps in the figure using numbered sequences.

**Reviewer Confidence:**

3: The reviewer is confident but not certain that the evaluation is correct

**Scope:**

4: The work is relevant to the Web and to the track, and is of broad interest to the community

---

### Official Review · Reviewer_pZxG · 2024-12-01

**Novelty:** 4
**Technical Quality:** 4

**Review:**

This paper proposes a model-agnostic social network refinement framework, which is primarily inspired by diffusion models. The framework effectively reduces the impact of noise inherent in social networks on the performance of social recommendation models. Overall, the authors' approach of leveraging diffusion models to address noisy information in domain-specific graphs is well-justified. They have effectively adapted and optimized the model for the social recommendation scenario. The incorporation of curriculum learning introduces a degree of novelty. However, the paper still has some shortcomings that could be further improved.

1. While I understand the author’s current organizational structure, I find the arrangement of the Introduction and Related Work sections somewhat lacking. Given the extensive research on social network refinement and the widespread application of diffusion models in areas like graph denoising, the authors should dedicate more space to exploring the limitations of existing methods from a mechanistic standpoint.

2. The three datasets used by the authors are limited in scale, which differs from real-world industrial scenarios. This limits the ability to assess the model's practical effectiveness. Furthermore, while the authors discuss the model's time complexity, they do not provide a comparison with existing methods. Moreover, including a comparison of runtime performance across varying data scales and experimental conditions would offer more meaningful insights.

3. The ablation study is incomplete; for instance, it remains unclear whether different difficulty calculation strategies in curriculum learning significantly impact model performance. Moreover, the paper lacks case studies of the model, particularly error analysis.

4. The paper does not provide the preprocessed datasets or code for the experiments, lacking reproducibility.

**Questions:**

Could you provide experimental results on any larger-scale dataset?

**Reviewer Confidence:**

3: The reviewer is confident but not certain that the evaluation is correct

**Scope:**

4: The work is relevant to the Web and to the track, and is of broad interest to the community

---

### Official Review · Reviewer_7yoQ · 2024-12-01

**Novelty:** 5
**Technical Quality:** 4

**Review:**

This paper presents ARD-SR, a model-agnostic social network refinement framework using diffusion models for more robust social recommendation (SR). This work aims to address important challenges in social recommender systems - the noisy social connections and limited social neighbors for many users, namely noise and sparsity.

Specifically, the authors present a method that leverages diffusion models to refine social networks by removing noisy connections and adding meaningful ones through a position-specific noise schedule in the forward process and a neighborhood preference-guided reverse process. The framework uses curriculum learning for training and alternates optimization between the diffusion model (DM) and SR backbone.

The proposed method is evaluated on 3 different SR backbones and 3 different real-world recommendation datasets, against 6 baselines.

Pros:
1. the paper is mostly well-written and well-organized
2. the curriculum learning strategy is reasonable. In using DM, the authors also design an interesting position-sepcific noise schedule based on user preference similarity and neighborhood preference guided reverse process
3. extensive experiments with detailed analysis of different user groups and statistical analysis of refined networks

Cons:
1. I personally do not reckon that the users have clearly demonstrated how close the social network refinement task aligns with the nature of DM. First, the fundamental nature of data differs, i.e., continuous pixels vs binary adjacency matrices. While the authors have mentioned this in the Introduction, the paper does not justify how the continuous Gaussian noise process meaningfully applies to binary edge existence. Second, in image generation, the corrupted intermediate states (during the forward process of DM) maintain some semantic meaning, while for social networks, it is unclear what a "partially noisy" connection represents.
2. I understand that the "position-specific noise schedule" is proposed as an effort to address the discrete nature of social networks. However, the resulting noisy states exist in continuous space, losing the binary semantic meaning of social connections still.
3. The paper uses preference similarity to scale noise levels, assuming higher similarity should lead to less noise. However, this creates a circular dependency: we use potentially noisy preference embeddings to determine how much to denoise, which is a bit contradictory to me.
4. The overall optimization is based on alternating training - no convergence analysis of this procedure is provided.
5. While a complexity analysis is provided, it seems somewhat superficial. In the sense of DM training, empirical scaling evidence is also expected, such as runtime comparison with varying network sizes and social network scales.

**Questions:**

1. The position-specific noise schedule is a key component, but is it really able to handle the gap between continuous and discrete data spaces?
2. How do you justify that the exponential decay function is the optimal choice for scaling noise based on preference similarity?
3. About the alternating training between ARD-SR and SR backbone: have you observed any instability issues?
4. How much computation overhead does incorporating DM into the training of the recommender system incur exactly?

**Reviewer Confidence:**

3: The reviewer is confident but not certain that the evaluation is correct

**Scope:**

4: The work is relevant to the Web and to the track, and is of broad interest to the community

---

### Official Review · Reviewer_MTDj · 2024-12-02

**Novelty:** 6
**Technical Quality:** 5

**Review:**

The paper is innovative in its application of diffusion models to address noise and sparsity in social recommendation systems, demonstrating originality and significance. It offers a well-structured and clear presentation of the ARD-SR framework, supported by comprehensive experiments across multiple datasets. However, certain sections could benefit from further simplification and explanation, and addressing minor formatting and efficiency-related discussions would enhance the overall quality.

Specifically, this paper introduces a model-agnostic social network refinement framework (ARD-SR) based on diffusion models to address two key challenges in social recommendation: noisy social connections and sparse user social neighbors. It incorporates several innovations, including position-specific noise scheduling for better noise modeling, neighbor preference-guided reverse processes for personalized network optimization, curriculum-based training to gradually handle complex samples, and joint training strategies for mutual enhancement between social network refinement and recommendation models. Experiments demonstrate that ARD-SR significantly improves recommendation accuracy and robustness and can seamlessly integrate with various recommendation backbones.

Pros:

1.	This paper introduces diffusion models to tackle the dual challenges of noise and sparsity in SRs.

2.	Curriculum-based training is proposed to strengthen denoising capability.

3.	Mutual enhancement strategy is designed for robust performance.

4.	Comprehensive experiments validate efficacy across multiple datasets and metrics.

Cons:

1.	In line 578, it is mentioned that the hyperparameter $\rho$ is controlled by a decay factor and will decay over time. However, there is no further discussion or explanation regarding this decay factor in the subsequent text.

2.	The hyperparameters for the noise bound, $\beta_{\text{min}}$ and $\beta_{\text{max}}$, are introduced but lack detailed discussion or analysis in Parameter Sensitivity Analysis. Further elaboration on their roles, impact on the model's performance, and guidelines for tuning these parameters would enhance the clarity and usability of the proposed framework.

3.	There is a lack of discussion comparing the efficiency of the proposed method with existing algorithms.

4. The endings of some equations are inconsistent, with some including punctuation and others not (e.g., equations 6, 7, and 22 lack punctuation). A unified formatting style is needed.

**Questions:**

1.	It is necessary to provide explanations or discussions regarding the aforementioned hyperparameters (decay factor, $\beta_{\text{min}}$ and $\beta_{\text{max}}$).

2.	When discussing the performance comparison across user groups with different numbers of social neighbors, only the Ciao dataset is considered. The results on the other two datasets could be reported in the appendix.

3.	Including a comparison of time overhead would make the content of the paper more comprehensive.

4.	Minor issues in the paper, such as spelling errors or inconsistent formatting, also need to be addressed.

**Reviewer Confidence:**

3: The reviewer is confident but not certain that the evaluation is correct

**Scope:**

4: The work is relevant to the Web and to the track, and is of broad interest to the community